# Joint Sentiment Part Topic Regression Model for Multimodal Analysis



**Mengyao Li, Yonghua Zhu \*, Wenjing Gao, Meng Cao and Shaoxiu Wang**

Shanghai Film Academy, Shanghai University, Shanghai 200444, China; limengyaoyao@shu.edu.cn (M.L.); gaowj715@shu.edu.cn (W.G.); cicimeng@shu.edu.cn (M.C.); iamalion@shu.edu.cn (S.W.)
\* Correspondence: zyh@shu.edu.cn

**Abstract:** The development of multimodal media compensates for the lack of information expression in a single modality and thus gradually becomes the main carrier of sentiment. In this situation, automatic assessment for sentiment information in multimodal contents is of increasing importance for many applications. To achieve this, we propose a joint sentiment part topic regression model (JSP) based on latent Dirichlet allocation (LDA), with a sentiment part, which effectively utilizes the complementary information between the modalities and strengthens the relationship between the sentiment layer and multimodal content. Specifically, a linear regression module is developed to share implicit variables between image–text pairs, so that one modality can predict the other. Moreover, a sentiment label layer is added to model the relationship between sentiment distribution parameters and multimodal contents. Experimental results on several datasets verify the feasibility of our proposed approach for multimodal sentiment analysis.

**Keywords:** sentiment analysis; multimodal; topic model

## 1. Introduction

With the development of social networks, the multimedia data that integrate texts and images, instead of single modality data, are gradually becoming the main medium for sentiment conveyance. Multimodal contents can make up for the lack of information in single modalities and enable sentiment expression that is more profound and mellifluous. Hence, more and more research attention has been shifted to multimodal sentiment analysis in the area of artificial intelligence. Automatic assessment of multimodal sentiment, which infers the sentiment information from digital contents from multiple modalities, will be of great significance in people's social and cultural life [1]. Besides applications in social networks, the research results can also be applied to many other fields, such as stock prediction, brand positioning, and medical resources [2–7].

Current approaches to multimodal sentiment analysis can be classified into three types [8–18]. The first is to predict the sentiment information given single-modality generated feature vectors as inputs, which ignores the information complementation between modalities and may cause trouble in processing an input of a large amount of redundant information [8–12]. Others attempt to make decisions based on the results of multiple sentiment classifiers, each of which is trained separately [13,14]. However, they are unable to effectively capture the correlation between the different modalities. Moreover, some researchers propose to carry out fusion in the middle layer of the whole model, mainly through neural networks. This kind of method merges connected multiple-feature units into a shared presentation layer. Although the methods based on intermediate-level fusion achieve good results in multimodal modal analysis, they fail in tackling incomplete multimodal contents.

To solve these problems, we propose a topic model which introduces a sentiment label (positive or negative) layer. Moreover, our model has two separate sets of multimodal feature processing

methods that are related by a linear regression module. Firstly, we extract eigenvectors of the input data from each modality. Secondly, two topic models are prepared to process the information of different modalities and apply the models' respective eigenvectors to sentiment analyses. In order to strengthen the inter-modal relationship, a linear regression module with latent variables is proposed. This module utilizes the complementary information between different modalities. Then a sentiment label layer is introduced to mine the model sentiments by using the distribution of sentiment topics and the relationship between features, so as to derive the final multimodal sentiment analysis results.

Our main contributions lie in the following three aspects:

We propose a joint sentiment part topic regression model (JSP). Our method can automatically generate the sentimental polarity of the document using the intermodal information and sentiment labels.

We add sentiment label layers to the framework to make the analysis results more reliable. A two modalities feature and sentimental labels are used to explore the internal relationship between modal and sentimental to solve the problem of multimodal content sentiment analysis.

The proposed model is tested on four real-world datasets of different orders of magnitude. The evaluation results demonstrate the effectiveness of our method in multimodal sentiment analyses. The experimental results show that our proposed model has great sentiment analysis ability in real-world multimodal datasets and has good performance when compared with the state-of-the-art sentiment recognition methods.

The rest of this paper is organized as follows. Section 2 is a summary of related works. Section 3 states the details of our proposed model. The mathematical inference is then elaborated in Section 4. The experimental setup and analysis of the results are presented in Section 5. We finally conclude this work in Section 6.

## 2. Related Works

Various kinds of approaches have been proposed for sentiment analysis. Our model aims to tackle multimodal sentiment analysis based on an extended latent Dirichlet allocation (LDA) model with a sentiment part. Hence, we focus on elaborating the related studies in both sentiment analysis and LDA models.

### 2.1. Single Modality Sentiment Analysis

#### 2.1.1. Textual Sentiment Analysis

One of the research hotspots in natural language processing (NLP) is sentiment analysis. There are three main methods: traditional machine learning [19–23], sentiment lexicographical methods [24–30], and deep learning [29–33].

In traditional machine learning, sentiment analysis was accomplished in the weak supervision environment by multi-dimensional continuous injection. Maas et al. drove a probabilistic model to capture semantic similarities among words, of which the component does not require marked data [20]. Pang et al. used adjectives in the text as features and tried to combine multiple features for sentiment analysis [23]. In the lexicon-based approach, a semantic orientation calculator (SO-CAL) was proposed to analyze textual sentiment by co-annotation of the polarity and strength of semantics [25]. Turney designed sentiment patterns based on word labels of two consecutive words. Phrases satisfying this pattern were extracted to obtain their semantic orientation [28]. With the increasing popularity of neural networks, many studies on sentiment analysis resort to deep learning. HAN made use of the recurrent neural network (RNN) model with a time series structure, which was a hierarchical cyclic neural network model based on an attention mechanism for text classification [33]. TextCNN for a text sentiment classification task was proposed. After the word vectors were pre-trained, Kim et al. used the convolutional neural network (CNN) to train the sentence-level classification task [32].

### 2.1.2. Visual Sentiment Analysis

Visual sentiment analysis is more challenging than textual sentiment analysis for the subjectivity of images. In general, visual sentiment analysis can be divided into three categories according to the features they utilize, low-level features [34,35], middle-level features [36,37], and high-level features [38,39] included. Zhao et al. proposed the method of sentiment feature extraction based on principles-of-art emotion feature (PAEF), which was to extract low-level visual features from the concept of art principle for image sentiment classification and scanning tasks [34]. In study of Borth et al., the adjective–noun pairs were detected to form mid-level features based on visual sentiment ontology [35]. In using high-level features, You et al. made an attention mechanism to automatically discover the relevant visual targets for image sentiment analysis [39]. In previous works, although the results of single modality analysis of sentiment perform well, this method cannot be applied to the multimodal content of social media. The single modality ignores the information complementarity between modalities and fails to make full use of the social media data. Moreover, due to the inadequate information, the sentiment analysis's robustness with single modality is weak. Hence, it is necessary to effectively process the multimodal sentiment analysis of diverse information.

### 2.2. Multimodal Sentiment Analysis

Multimodal contents have gradually become the main carrier of sentiment information in the social networks. Various kinds of approaches have been proposed for multimodal sentiment analysis, most of which focus on how to effectively integrate information of different modalities from the feature or decision levels. Feature-level fusion attempts to take the fused features of different modalities as input for the sentiment classification model. Poria et al. used a deep CNN to capture multimodal features, which were then integrated by a multiple-kernel learning sentiment analysis classifier [14]. In order to capture contextual information in different modalities, a cyclic neural network was used to integrate features in pairs [15]. A utilized attention mechanism was used to guide the feature selection of single-modality information for multimodal feature fusion [16,17]. Decision-layer fusion is also a widely used approach to multimodal information fusion, such as Kalman filters [18], support vector regression [19], and Bayesian inference [20]. That is, a specific model is adopted to extract single modality features for different data types, and the complex relationship between different modality prediction results is modeled through different fusion methods. At present, most of the multimodal sentiment analysis only focuses on sentiment classification, and does not pay attention to the topic, which limits the depth of user validity. Second, most of these studies rely on a tagging and training-related corpus, leading to limitations in other interest areas of the target tasks. It can be said that most analytical techniques ignore the complex and changeable relationship between multimodal information and sentiment.

### 2.3. Multimodal Latent Dirichlet Allocation

With the development of multimodal analysis, many studies have focused on the use of a topic model like LDA, which can effectively model documents. Blei et al. proposed multimodal-LDA (mm-LDA) which introduced a group of shared latent variables to obtain the correlation between two modalities, so that the two modalities shared topic proportion variables [40]. They further extended mm-LDA to correspondence LDA (cLDA), in which each caption word shares a hidden topic variable with a random image [41]. Hagai et al. proposed topic regression multimodal LDA (tr-mmLDA) which can learn two independent latent variables independently [42]. A regression module was then introduced to allow a topic to predict another in a linear way. Besides, their calculation formulas allowed for the captured data to be different, which means that the latent topic numbers of the image and text can be different. Meanwhile, the LDA model can be used in global or individual topic generation [43] and is suitable for comment-oriented modeling. For example, Lin et al. added a connection between the topic and the sentiment on the basis of mm-LDA, and the new model can

extract and summarize relevant sentimental text [44]. Ding et al. proposed the maximum entropy LDA (me-LDA), which combined the topic model with the maximum entropy [45].

Different from the above-mentioned works, we introduce linear regression modules to establish inter-modal relationships, which can utilize the complementary information well at the same time. Then, we innovatively add a sentiment label layer to associate the sentiment topic distribution with fused multimodal features.

## 3. Method

The proposed topic model with a joint sentiment part (JSP) is based on latent Dirichlet allocation (LDA) [46]. Here, we give a brief introduction to the LDA model. This topic model has a concise internal logical structure and a series of complete mathematical inference algorithms. Its mathematical inference process utilizes Bayes' theorem, input prior knowledge, and multimodal data, where the prior knowledge is processed in Section 5.3. LDA is a generation probability model based on a three-layer Bayesian network, and each layer can be adjusted by relevant parameters, which also provides the flexible scalability of the model and can effectively manage the diversified multimodal content. Based on the LDA model, the input image or text can be transformed into a topic-based representation, which reduces the feature dimension and facilitates the multimodal fusion.

Its framework is shown in Figure 1. LDA is a variety of generation model that can hierarchically model words or images of documents to be analyzed. As shown in Figure 1, there are two latent variables in LDA. The first one is the topic distribution $z_n$ which samples the current word to topic T. The other one is $\theta_d$ which is the relationship between the text d and each topic from the Dirichlet distribution with the parameter $\alpha$.

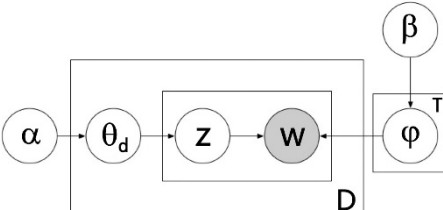

**Figure 1.** Bayesian network of latent Dirichlet allocation (LDA) [47].

Specifically, LDA represents the distribution of words in the document as the contribution of T topics, and models the topic proportion as the Dirichlet distribution, where each topic is a multinomial distribution of words. The joint probability distribution of all words in a text and their topics is:

$$P(w, \ z|\alpha, \ \beta) = P \ (w|z, \ \beta) \ P(z|\alpha), \tag{1}$$

$$\int P(z|\theta)P(\theta|\alpha)d\theta = \int P \ (w|z, \ \varphi) \ P(\varphi|\beta) \ d\varphi. \tag{2}$$

### 3.1. Proposed Method

#### 3.1.1. The Overview of Proposed Method

People are used to sharing their opinions on social media. Therefore, refining the topic of the viewpoint can achieve the purpose of analyzing its sentimental direction. For multimodal sentiment analysis, we introduce a topic model to process multimodal data. For tackling multimodal data, traditional LDA methods ignore the deep relationship between the two modalities. So, we follow tr-mmLDA to address the problem of finding shared latent variables between data modalities. Figure 2 shows the structure of our proposed model. In detail, the model is constructed by an additional module with parameter x to make a linear influence on the relationship between visual vectors and textual vectors. Besides, this linear module is a Gaussian regression module, which takes the proportion of

the textual topic z as input, and the proportion x of the visual hidden topics in the annotated textual vector as a response variable. Two separate topic models are used to generate a visual word w and the textual word r. Traditional LDA multimodal sentiment analysis methods do not take the hidden relationship between the sentiment and the document topic into account, so we add a sentiment label layer to deepen the connection between the document and topic layer. In this way, sentimental label l is linked to textual topic z, label l can influence the image vector through the linear module at the same time. Figure 3 gives an example of predicted values assigned. Finally, we derive the model using variational expectation maximization (EM).

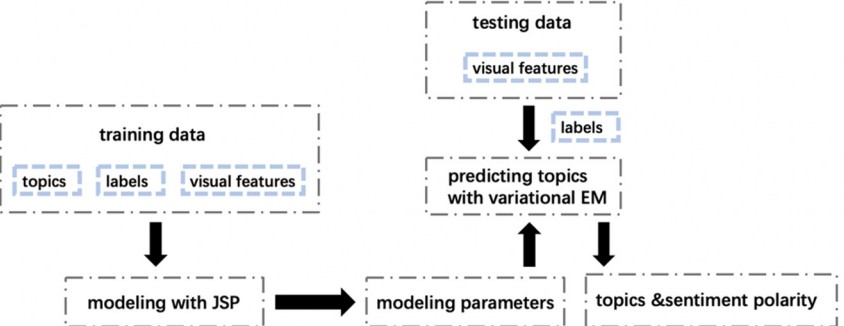

**Figure 2.** The structure of proposed method.

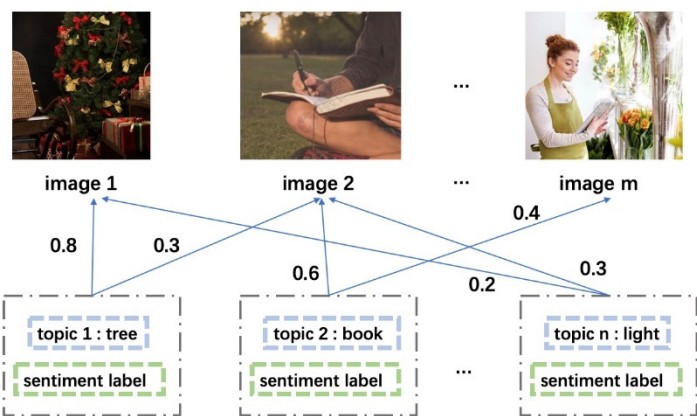

**Figure 3.** The example of predicted values assigned.

### 3.1.2. Data Representation

From successful works of scene classification, we chose to use the bag-of-word (BoW) model [22] to represent images and text. The BoW model is understandable and efficient so it is widely used. First appearing in the field of natural language processing and information retrieval, BoW is used to process the identification and classification of documents [21]. Later, it was widely used in image target classification and scene classification [7]. In this way, document data are converted into a simple result of word vectors without any consideration for word order. Meanwhile, the BoW model is used to simulate the image as a document, and the local features in the image are extracted as visual words. Therefore, the multimodal content is reduced to a pair of total word vectors, where the basis vector of the visual word is defined as the unit vector $v \in \mathbb{R}^{1 \times U_v}$, and similarly, the basis vector $t \in \mathbb{R}^{1 \times U_t}$ is also defined. Thus, $V = \{v_1, v_2, \ldots, v_m\}$ can represent an image that is constructed of $m$ words, and $T = \{t_1, t_2, \ldots, t_n\}$ can represent a text including $n$ words.

### 3.1.3. Joint Sentiment Part Topic Regression Model (JSP)

Our proposed topic model is shown in Figure 4. We suppose the corpus with *D* documents, given *L* visual topics and *K* textual topics. Then, we assume that *N* text features and *M* image features are extracted from the document.

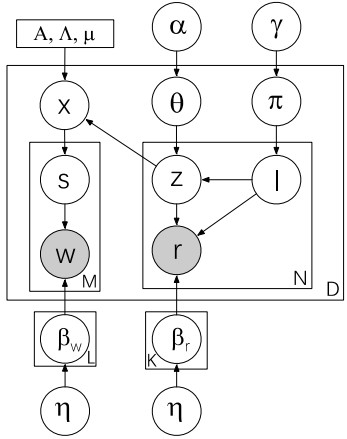

**Figure 4.** The diagram of proposed method.

The text hidden topic is denoted as $Z = \{z_1, z_2, \ldots, z_n\}$, and the given topic proportion as $\theta$. Meanwhile, the visual parameter *x* is obtained by a set of linear regression formulas involving *Z*. Here, we follow [47]:

$$x = A\bar{z} + \mu + n, \tag{3}$$

where A is a $K \times L$ regression coefficient matrix. The value of $\bar{z}$ is borrowed from previous works on supervised LDA [4,14]:

$$\bar{z} = \frac{1}{N} \sum_n z_n. \tag{4}$$

At this point, the visual word and the textual word are connected through the linear regression module that involves parameter *x*.

Sentiment topic distribution $\pi$ is extracted from the sentiment Dirichlet distribution with parameter $\gamma$. After that, the sentiment label *l* is selected from the multinomial distribution of sentiment topic distribution $\pi$. Meanwhile, the parameter $\beta$ ($\beta_w$ and $\beta_r$) which is the relationship that between topic and word is extracted from the Dirichlet distribution with parameter $\eta$.

Furthermore, we sample the proportion $\theta_{d, l}$ of each sentiment label *l* under document *d* from the Dirichlet distribution with parameter $\alpha$. When there are *K* topics, $\theta$ is a k-dimensional vector, and each element represents the probability of the topic appearing in the document. After that, the current feature belongs to the topic that is sampled from the multinomial distribution with parameter $\theta_{d, l}$. Finally, the word distribution based on topic and sentiment label is extracted from the multinomial distribution with parameter $\beta_{Z_{d_n,l}}$, and the textual word *r* in the document is extracted based on the multinomial word distribution.

The definition of the processing process of image–text pairs containing M image feature words and N text words is as follows:

- For the sentiment label l:

    - Select an assignment $\pi \mid \gamma \sim$ Dir ($\gamma$);
    - Find the sentiment label $l|\pi \sim$Mult($\pi$).

- For each text word r∈ {1, . . . , N}:

    - Find a topic distribution $\theta \mid \alpha \sim$Dir ($\alpha$) and a word assignment $\beta|\eta\sim$Dir ($\eta$);

- - Find a topic z in document d, and z~Mult ($\theta_{d,l}$);
  - Figure out the real text word r with the parameter $\beta_{z_{d_{n,l}}}$ from the multinomial distribution.

- Introduce the mean Gaussian linear regression parameter x:

  - Let the proportion of x be: $x = A\bar{z} + \mu + n$

- For the image feature w:

  - Choose a topic distribution s|x~Mult(x);
  - Choose the image feature w with the parameter $\beta_{S_{X, M}}$ from the multinomial distribution.

### 3.1.4. Model Optimization

In the regression module, the stronger the correlation between the data of the two modalities is, the larger the coefficient of A in the module is. Hidden topics with multiple texts can add impact factors to the corresponding images. By minimizing the value of π□, $\theta$, and β ($\beta_w$ and $\beta_r$), there will be a stronger connection between the labeled text and the images. Our model has greater flexibility in capturing the relevance of content. Therefore, this model also requires relatively low expression forms.

## 4. Mathematical Inference

To find the distributions of $\pi$ ☉ , $\theta$ and β ($\beta_w$ and $\beta_r$), we need to infer the posterior distribution of the parameters z and l, which means finding the word distribution for the topic and sentiment labels. Here, we use the external datasets MDS [48], Subject MR [49], and Review Text Content [50] as prior knowledge. Let P (z, l) be the sampling distribution of the words given the remaining topics and sentimental labels. It can be obtained that the joint probability between the text, topic, and corresponding sentimental label distribution in the document can be decomposed as follows:

$$P(\mathbf{r,\ z,l}) = P(r|z,l)P(z,l) = P(r|z,l)P(z|l)P(l), \tag{5}$$

for the P(r, z, l) of the above formula, we can integrate $\beta$, and get:

$$P\ (\mathbf{r|z,l}) = \left(\frac{\Gamma(V\eta)}{\Gamma(\eta)^V}\right)^{K\times T} \prod_k \prod_j \frac{\prod_i \Gamma(N_{k,j,i} + \eta)}{\Gamma(N_{k,j} + V\eta)}. \tag{6}$$

the $N_{k,\,j,\,i}$ in the formula represents the number of times the word i appears in both the topic j and the sentiment label k. $N_{k,\,j}$ represents the number of occurrences of the word in the distribution of the topic j and the sentiment label k. There is also a gamma function Γ [51,52].

Second, the method of integration to $\theta$ can be used to obtain:

$$P\ (\mathbf{z|l}) = \left(\frac{\Gamma\left(\sum_{j=1}^{T} \alpha_{k,j}\right)}{\prod_{j=1}^{T} \Gamma(\alpha_{k,j})}\right)^{D\times K} \prod_d \prod_k \frac{\prod_j \Gamma(N_{d,k,j} + \alpha_{k,j})}{\Gamma(N_{d,k} + \sum_j \alpha_{k,j})}. \tag{7}$$

Next, the method of integration to $\pi$ can be used to obtain:

$$P\ (\mathbf{l}) = \left(\frac{\Gamma(K\gamma)}{\Gamma(\gamma)^K}\right)^{D} \prod_d \frac{\prod_k \Gamma(N_{d,k} + \gamma)}{\Gamma(N_d + K\gamma)}. \tag{8}$$

The parameter x of the reasoning image feature part is what we draw on from previous research. Assuming the latent topic in the text feature is z, the distribution x between the image topic and the image feature word is $x = A \underset{z}{\rightarrow} + \mu + n$, of which A is the K ×L dimensional matrix, $\mu$ is the vector of the average parameter, and n is the introduced Gaussian constant. LDA in a supervised

environment [4,14] makes $\underset{Z}{\rightarrow} = \frac{z_1+z_2+...+z_N}{N}$. After this calculation, the input of the added linear regression module will affect the generation of image features, so that the topic distribution ratio of image features is affected by the text words. According to $P(l|d)$, which is the probability of the sentimental label l of a given content d, the multimodal content can be sentimentally classified.

## 5. Experiments

In the experimental part, a series of experiments are carried out to confirm the validity of the proposed model for multi-modal sentiment analysis. Specifically, we compare the model with some baseline models and the variant of our model on several real-world datasets qualitatively. The selection of parameters in the model is also discussed. However, the previous work has demand in the marked sentimental datasets, which results in the inaccuracy of the multimodal content analysis. This paper accepts the binary classification that only considers the probability of positive and negative sentimental labels in the object content to limit the above drawback. The method we proposed makes great use of the available sentimental datasets to analyze multimodal content. Besides, the classification of labels in the sentiment dataset we use is a binary classification problem, the prior information set only affects the positive and negative words.

### 5.1. Datasets

- Flickr [53]: Flickr is one of the most commonly used multimodal sentiment datasets and contains 297,462 image–text pairs with weak markers, of which 143,542 datapoints are labeled negative. It contains 1211 adjective–noun pairs. The corresponding image of adjective-noun pairs (ANPs) can be queried through its API. The sentimental labels of these image–text pairs are also marked by the corresponding ANP.
- Flickr-ML [54]: Jie et al. [53] constructed Flickr-ML which contains 21,037 image–text pairs, of which 10,145 have negative markers. This dataset is partially labeled with sentiment and has a more accurate classification label. Jie et al. randomly selected 30,000 image–text pairs in the original Flickr, split evenly between positive and negative tags. Then, five annotators were used to mark the image–text pairs to calibrate the sentiment labels. In this way, the strongly tagged Flickr-ML dataset was obtained.
- Twitter [55]: 24,795 image–text pairs were obtained after filtering the original 183,456 pieces of data crawled by a relevant API. In Twitter, there are a total number of 12,357 negative pairs.
- Visual sentiment ontology (VSO) [56]: This dataset contains a total of 603 pictures, which are split into 21 types. Ziyuan Zhao et al. used the processing method to extract the middle layer visual features to obtain the similarity coefficient, and the data were weakly labeled by support vector machine (SVM). We screened out 564 qualified data, including 276 negative data. Among them, 400 pieces of data are selected use in training, and the rest as the test data of the later model performance.

  Sentimental label datasets:

- Review Text Content (RTC) [57]: This dataset is composed of restaurant reviews collected and manually annotated by Ganu et al. It contains 150 different restaurants and more than 3200 sentences, which are broken down into six broad categories and each sentence contains one or more categories of labels.
- Subject MR (subjMR) [58]: The subjective MR used here is a new version of Pang and Lee et al., which screens out data containing opinion information based on the original MR dataset, including more than 2000 documents.
- MDS (Multi-domain Sentiment) [59]: This dataset is built by Blitzer et al. and collected from product reviews on Amazon, including electronics, kitchenware, DVDs, and books. There are more than 2000 pieces of data in each category, with positive and negative labels accounting for half.

*5.2. Baseline Models*

In order to explore the performance of our proposed method, we also compared our model with the following state-of-the-art methods.

- Text only [60]: The model uses text features through a logistic regression model. The text topic is extracted and then the sentiment is classified by SVM.
- Visual only [61]: Just uses deep visual features through a logistic regression model.
- CCR [62]: A cross-modality consistent regression model, which uses progressive CNN to extract image feature and title information to represent the text information, for joint textual–visual sentiment analysis.
- JST [54]: Joint sentiment topic model, a probabilistic model framework based on LDA, this model can infer both sentiments and topics in the documents. Different from JS-mmLDA, JST is a single-modality sentiment analysis model, which is only used to process textual information.
- T-LSTM-E [62]: Tree-structured long short-term memory embedding, an image–text joint sentiment analysis model which integrates tree-structured long short-term memory (T-LSTM) with visual attention mechanism to capture the correlation between the image and the text.
- TFN (Tensor Fusion Network) [63]: A deep multimodal sentiment analysis method modeling intra-modality and inter-modality dynamics together into a joint framework.

*5.3. Preprocessing*

Before we conduct the experiments, we do some preprocessing on the input data. Figure 5 shows the processing of the input data. For better fitting of the model, the text length is kept to no less than six words and no more than 80 words. In detail, the size of the images in the dataset is clipped to 256 × 256 pixels. Without special instructions, 70% of the data in each category are used for model training, and the remaining 30% are used for performance tests of later models. For textual contents, numbers, stop words, punctuation marks, and some non-alphabetic characters are removed. Following the previous work, the prior knowledge utilized in our model is combined with two subjective lexicons, namely, MPQA (multi-perspective question answering) and the evaluation lexicons. The words in both lexicons are marked with sentiment polarity. After selecting the words with strong polarity, we carried out word stem extraction. Among them, 4196 words were extracted, including 2427 negative polar words. The final prior information is generated by retaining all the words in MPQA and evaluation lexicons that appear in the experimental dataset. Besides, by performing standard word stem analysis, the vocabulary overfitting can be solved.

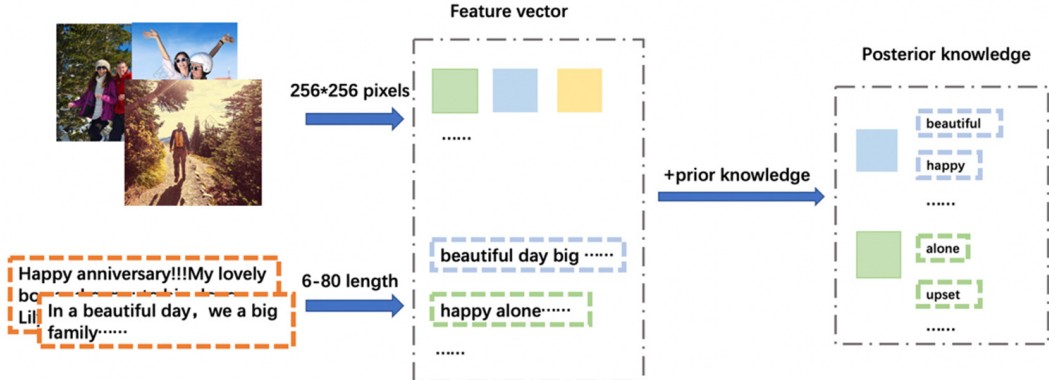

**Figure 5.** The processing of the input data.

*5.4. Experiment Evaluation Metrics*

To evaluate the performance of the model intuitively and analyze the experimental data quantitatively, we used these four metrics to evaluate the results of the experiment, namely, recall, accuracy, F-score, and precision. These four metrics are defined as follows:

$$\text{Recall} = TP/(TP + FN), \tag{9}$$

$$\text{Accuracy} = (TP + TN)/(TP + TN + FP + FN), \tag{10}$$

$$\text{F-score} = (2 \cdot \text{Precision} \cdot \text{Recall})/(\text{Precision} + \text{Recall}), \tag{11}$$

$$\text{Precision} = TP/(TP + FP). \tag{12}$$

*5.5. Experiment Results and Analysis*

5.5.1. Comparative Experiments with Different Datasets

In the subsequent experiments, we analyze the results through four different datasets by four indicators: recall, accuracy, F-score, and precision.

The experimental results in Tables 1–4 directly show the comparison between our proposed model with other baselines in four datasets. It can be seen that the models including text only and images only are not well behaved as these two models only take unimodal data into account. The other models, which considered multiple modalities or joint sentiment labels and treated them appropriately, performed better than single-modality models.

**Table 1.** Performance of various sentiment analysis methods on Flickr.

| Method | Recall | Accuracy | F-Score | Precision |
|---|---|---|---|---|
| Proposed method | 0.854 | 0.847 | 0.819 | 0.856 |
| Text only | 0.734 | 0.722 | 0.694 | 0.674 |
| Visual only | 0.742 | 0.732 | 0.712 | 0.693 |
| CCR (cross-modality consistent regression model) | 0.823 | 0.822 | 0.821 | 0.832 |
| JST (joint sentiment topic mode) | 0.844 | 0.817 | 0.823 | 0.832 |
| T-LSTM-E (tree-structured long short-term memory embedding) | 0.851 | 0.844 | 0.83 | 0.843 |
| TFN (tensor fusion network) | 0.867 | 0.832 | 0.832 | 0.844 |

**Table 2.** Performance of various sentiment analysis methods on Flickr-ML.

| Method | Recall | Accuracy | F-Score | Precision |
|---|---|---|---|---|
| Proposed method | 0.874 | 0.865 | 0.827 | 0.864 |
| Text only | 0.769 | 0.776 | 0.743 | 0.722 |
| Visual only | 0.821 | 0.789 | 0.752 | 0.743 |
| CCR | 0.816 | 0.838 | 0.827 | 0.837 |
| JST | 0.826 | 0.834 | 0.826 | 0.843 |
| T-LSTM-E | 0.862 | 0.857 | 0.833 | 0.862 |
| TFN | 0.882 | 0.863 | 0.852 | 0.867 |

Tables 1 and 2 show the results for large datasets. The operational results of JST are significantly improved over the single feature model, as the JST model adds an additional sentiment layer based on the LDA. In this way, establishing a relationship between sentiment labels and feature vectors does improve the quality of sentiment analysis. T-LSTM-E improves by nearly 14% in recall compared with the single modality method, which is joint visual–textual semantic at the same time. The joint semantic embedding extracts the deep features between image–text pairs. The proposed model utilizes complementary information between modalities and joint sentimental labels to influence the distribution of document topics. Hence, compared with other state-of-the-art baseline methods,

our model has a good performance in precision. Our method improves by 2% in accuracy which also proves that the proposed model can accurately deal with the classification of multimodal sentiments. Flickr-ML is the dataset that has been manually labeled with sentiment tags. As shown in Table 2, all model performances are improved. The reason may be that the labels of the image–text pairs are marked as conforming to the corresponding sentiment keywords in the dataset with weak labels, which improves the effectiveness of the multimodal content analysis. As shown in Table 2, TFN obtains advantageous results by modeling dynamic relationships. Our proposed approach improves by almost 3% in precision, which can also validate the benefits of strong label handling.

**Table 3.** Performance of various sentiment analysis methods on Twitter.

| Method | Recall | Accuracy | F-Score | Precision |
|---|---|---|---|---|
| Proposed method | 0.859 | 0.842 | 0.836 | 0.843 |
| Text only | 0.711 | 0.721 | 0.685 | 0.687 |
| Visual only | 0.734 | 0.724 | 0.71 | 0.704 |
| CCR | 0.824 | 0.83 | 0.819 | 0.847 |
| JST | 0.832 | 0.823 | 0.814 | 0.833 |
| T-LSTM-E | 0.833 | 0.835 | 0.843 | 0.863 |
| TFN | 0.855 | 0.843 | 0.833 | 0.857 |

**Table 4.** Performance of various sentiment analysis methods on Flickr-ML.

| Method | Recall | Accuracy | F-Score | Precision |
|---|---|---|---|---|
| Proposed method | 0.843 | 0.842 | 0.869 | 0.865 |
| Text only | 0.725 | 0.764 | 0.724 | 0.711 |
| Visual only | 0.756 | 0.774 | 0.745 | 0.732 |
| CCR | 0.823 | 0.822 | 0.842 | 0.853 |
| JST | 0.832 | 0.834 | 0.848 | 0.859 |
| T-LSTM-E | 0.851 | 0.833 | 0.853 | 0.872 |
| TFN | 0.866 | 0.84 | 0.877 | 0.888 |

Tables 3 and 4 show the experimental results on small datasets like Twitter and VSO. Comparing experiments on other databases, it can be concluded that data volume does have an impact on the multimodal sentiment analysis task. As shown in Table 4, compared with single-modality models, our method is greatly improved in each evaluate metric, which indicates that information complementarity of multimodal content can capture and analyze sentimental content in a better way. From the results, our proposed model is as satisfactory as the state-of-the-art approach in some metrics. Besides, its learning ability and classification ability are better than other baseline models.

### 5.5.2. Ablation Experiments

In order to explore whether the sentiment label layer can effectively improve the ability of the model for analyzing the sentiment of multimodal contents, we construct a variant of our proposed model without a sentiment layer, named the no-sentiment layer model. Different from the proposed model which directly adds the sentiment label layer and analyzes the sentiment by using the relationship with the multimodal content, the no-sentiment layer model iteratively optimizes the model's latent variable parameters using variational expectation maximization. Through the variational EM, an approximate reasoning algorithm is used to replace the exact reasoning in the original E step. The new parameter estimation is obtained by calculating the strict lower likelihood bound in the E step of variation and maximizing the lower likelihood bound in the M (maximization) step. In this way, after the topic of each document is generated, the sentiments of each document are analyzed by SVM.

As shown in Figure 6a,b, the performance of the two models is compared for Flickr and Twitter respectively. Clearly, the model with the sentiment layer added to both datasets performed better, with the addition of the sentiment distribution resulting in a nearly 10% improvement in model

performance across all metrics. In particular, in the aspect of accuracy and precision are improved, thus establishing that the relationship between the sentimental layer and the multimodal content is helpful to improve the analysis accuracy of the model.

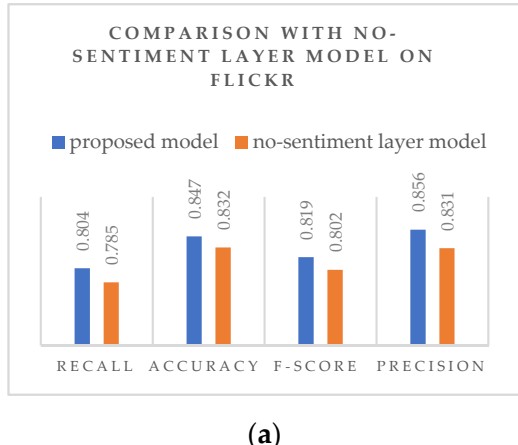

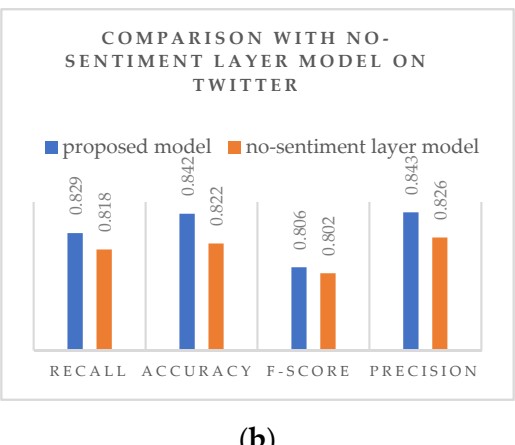

(**a**)        (**b**)

**Figure 6.** (**a**) Comparison experiment on Flickr; (**b**) comparison experiment on Twitter.

### 5.5.3. Experiments with Different Parameter Settings

The performance of multimodal sentiment analysis in our model also depends on the selection of parameter $\gamma$ and parameter $\eta$. Previous studies [53] have shown that LDA can generate good results by using symmetric prior knowledge settings. However, for document topic distribution types, the experimental results obtained by asymmetric prior knowledge are better than the former. In order to prove that parameters $\gamma$ and $\eta$ will affect the performance of the proposed model, we set up a comparison experiment on different parameter values in the experiment. Finally, in our proposed model, parameters are set based on the study of Griffith et al. [62], that is, $\gamma = (0.06 \times C) \div T$, where C is the average of document length and T represents the sum of sentimental labels, parameter $\eta = 0.02$. The assignment of the super parameter $\alpha$ can be obtained and iterated directly by using the maximum likelihood estimation method [63].

A common indicator is adopted here to measure the parameter performance: perplexity. Perplexity is usually used to measure the probability distribution or the quality of a probability model's prediction sample, that is, to estimate the quality of a probability model. It is calculated by the following formula:

$$P(p) = 2^{H(p)} = 2^{-\sum_x p(x) \log_2 p(x)}. \tag{13}$$

In general, the larger the perplexity value, the worse the result. While controlling the other variables, we conduct the adjustment experiments of a certain parameter. We expect to get a better parameter setting, making the perplexity value smaller. First, we fix the value of $\gamma$ and study the influence of parameter $\eta$ on perplexity. The broken line diagram is shown in Figure 7. Although the operation result is not perfect, we can see that the overall trend of broken lines reaches to the lowest point when $\eta$ takes the value of 0.02–0.03, and the lowest point is also within this range when $\gamma$ is given different values. So, we set the value of parameter $\eta$ to 0.02.

Similarly, it can be seen from Figure 8 that while exploring the effects of parameter $\gamma$ on perplexity, the value of parameter $\eta$. is kept unchanged, and we expect to find a minimum value to make the result of the model the best. At this point, the effect of $\gamma$ between 0.06 to 0.07 is better. The different values of $\eta$ show that the lowest point of the poly line is also within this range. This is why we chose to set the value of $\gamma$ to 0.06.

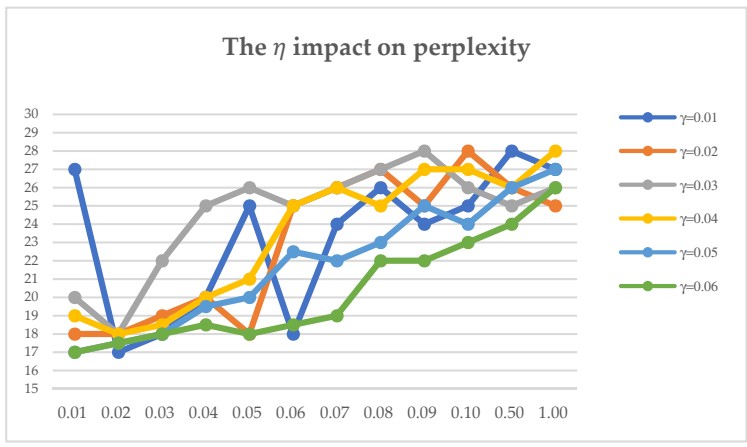

**Figure 7.** The $\eta$ impact on perplexity.

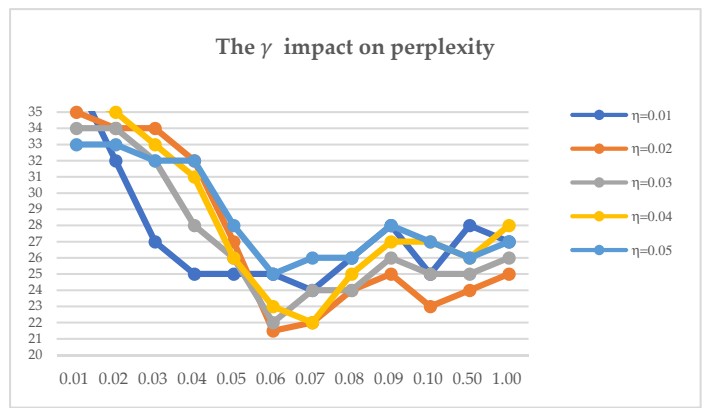

**Figure 8.** The $\gamma$ impact on perplexity.

## 6. Conclusions

In this paper, we propose a joint sentiment part topic regression model for multimodal sentiment analysis. Different from other LDA models that process and analyze multimodal contents, we introduce the sentiment label layer when the documents are generated. Not only is the corresponding relationship between the text and the image taken into account, but also the corresponding sentiment labels are related to document topics. The method we use allows for different numbers of image and text topics to be captured. Experimental results on different datasets like Flickr and Twitter also show the effectiveness of our proposed method in sentiment analysis of multimodal data.

**Author Contributions:** Conceptualization, M.L.; methodology, M.L.; software, M.L. and W.G.; formal analysis, M.L.; investigation, M.L.; resources, M.L. and W.G.; data curation, M.L. and S.W.; writing—original draft preparation, M.L.; writing—review and editing, M.C. and W.G.; visualization, M.L.; supervision, M.L. and Y.Z.; project administration, Y.Z.; funding acquisition, M.L. All authors have read and agreed to the published version of the manuscript.

**Funding:** This research was funded by the National Key Research and Development Plan of China, grant number 2017YFD0400101.

**Conflicts of Interest:** The authors declare no conflict of interest.

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
