# Peer review of "Joint Sentiment Part Topic Regression Model for Multimodal Analysis"

_information, doi:10.3390/info11100486_

Round 1

Reviewer 1 Report

Summary:
In the paper with the title "Joint Sentiment Part Topic Regression Model for Multimodal Analysis", an approach is proposed
to figure out the sentiment on data of large data sets based on multimodal considerations. The authors describe relevant
background information and discuss the related work very extensively. Then, the formal foundations of the approach are explained.
Based on this, experiments were conducted on several data sets. The results of the experiments are well-explained, and after that,
the authors conclude their results.

Point in favor:
- The paper is written very well
- The paper fits to the scope of the paper
- The paper discusses related works extensively and well
- The paper is sound
- The paper shows experimental results
- The paper deals with a topical subject
- The paper makes its contribution very clear
- The paper shows the formal foundations very well

Points againts the paper:
- Limitations must be discussed explicitly
- The results must be summarized in a short discussion
- More technical information about the used toolsets would be good
- Some minor language issues can be corrected, e.g.,
(1) First sentence of abstract sounds awkward
(2) Line 50, Page 2: part(JSP) -> part (JSP)
(3) Line 139, Page 3: Allocation(LDA) -> Allocation (LDA)
(4) Line 141, Page 4: algorithm -> algorithms
(5) Line 346: Table 3 - 4 -> Tables 3 - 4

Reviewer 2 Report

In this manuscript, Li et al. propose a Latent Dirichlet Allocation-based method with a regression module for sentiment analysis. Based on the thorough benchmark the authors provide, the presented hybrid method gives a significant improvement compared to the existing methods and their approach may become a real alternative to e.g. traditional machine learning or even deep learning models.  

  • My main concern is that the manuscript including the introduction and the results sections are difficult to understand as currently presented. I strongly suggest putting figures describing the nature and the structure of the data and models used. This would make the study (more) accessible to readers not familiar with these data sets. For example, the section 5.3 could use a figure explaining the model architecture better.
  • My other concern to this work, and in a broader sense to the whole field, is the interpretability. Machine learning models are one of the most amazing tools we use to analyse and interpret data sets from various domains. Many of them are ab ovo black box models, however there is a big effort to devise interpretable architectures. The authors should provide examples and show how certain entities are classified or the predicted values assigned. By doing so, the readership could get an insight into what exactly the developed model has actually learnt from the data. NLP should be a perfect domain to showcase such patterns inferred by the models.

Round 2

Reviewer 2 Report

The authors have taken my suggestions and, in my opinion, the manuscript has become eligible for publishing.
